# Incorporating Probing Signals into Multimodal Machine Translation via Visual Question-Answering Pairs

**Yuxin Zuo**[1†*]**, Bei Li**[2†]**, Chuanhao Lv**[2]**, Tong Zheng**[2]**,**
**Tong Xiao**[2,3‡] **and Jingbo Zhu**[2,3]

[1]School of Software, Northeastern University, Shenyang, China
[2]School of Computer Science and Engineering, Northeastern University, Shenyang, China
[3]NiuTrans Research, Shenyang, China
truman.yx.zuo@gmail.com,{libei_neu,lch-sy}@outlook.com
zhengtong12356@gmail.com,{xiaotong,zhujingbo}@mail.neu.edu.cn

## Abstract

This paper presents an in-depth study of multi-modal machine translation (MMT), examining the prevailing understanding that MMT systems exhibit decreased sensitivity to visual information when text inputs are complete. Instead, we attribute this phenomenon to insufficient cross-modal interaction, rather than image information redundancy. A novel approach is proposed to generate parallel Visual Question-Answering (VQA) style pairs from the source text, fostering more robust cross-modal interaction. Using Large Language Models (LLMs), we explicitly model the probing signal in MMT to convert it into VQA-style data to create the Multi30K-VQA dataset. An MMT-VQA multitask learning framework is introduced to incorporate explicit probing signals from the dataset into the MMT training process. Experimental results on two widely-used benchmarks demonstrate the effectiveness of this novel approach. Our code and data would be available at: https://github.com/libeineu/MMT-VQA.

## 1 Introduction

Broadening the scope of traditional machine translation, multimodal machine translation (MMT) enhances the quality of text translation by factoring in an auxiliary visual modality (Specia et al., 2016). The key challenge of MMT is finding an effective way to integrate images into the translation model, thereby maximizing the utilization of visual information (Caglayan et al., 2016; Libovický and Helcl, 2017; Calixto and Liu, 2017; Qian et al., 2018). Meanwhile, it operates on the premise that the relevant visual context can help clarify or fill in gaps when the source sentence is unclear or incomplete.

While, researchers soon discovered that MMT systems do not always perform as expected: the visual component often contributes little to the trans-lation process when the text is complete. Surprisingly, MMT systems can still behave well even the image input is unrelated to the text (Grönroos et al., 2018; Lala et al., 2018; Elliott, 2018). This raises questions about the actual role of visual input in the MMT framework. Aiming this, Caglayan et al. (2019) pointed out that vision features are helpful when the source sentence miss important patterns. Along this line, Li et al. (2022a) designed more specific probing tasks to evaluate how MMT behave in a limited textual context. But a severe problem still remains, that MMT systems exhibit less sensitivity to the information conveyed by the vision modality when the text is complete.

Our hypothesis posits that this may not result from the redundancy of visual data but rather from the lack of effective interaction between the text and visual modalities. It is a reasonable progression to consider the probing task proposed by Li et al. (2022a), which explicitly measures the cross-modality ability. This task involves masking critical entities and evaluating whether MMT can produce accurate translation by attending to the image.

However, this process falls short in providing any actionable feedback to MMT as it serves as a evaluation metric. Consequently, it is worth investigating whether we can incorporate the probing signals into the training process, instead of using it solely for evaluation.

In this study, we focus on augmenting MMT systems by incorporating probing signals during training to fortify the interplay between text and visual data. This can actively stimulating the textual representation to engage with visual data. To achieve this goal, we leverage Large Language Models (LLMs) to identify intricate contextual words and generate questions, thereby transforming original text into Visual Question-Answering (VQA) style pairs. This results in a new Multi30K-VQA dataset. On this basis, we introduce a MMT-VQA multi-task learning framework, prompting

---

[*]Work done during an internship at NiuTrans Research.
[†]Equal Contribution.
[‡]Corresponding author.

the MMT model to actively "probe" visual data guided by the textual context.

Our main contributions can be summarized as follows:

- We first demonstrate that utilizing advanced pre-trained vision features, such as MAE and BLIP, yields consistent performance improvements across various scenarios.

- We release a Multi30K-VQA dataset via using LLMs API, which consists of question-answering pairs originated from the source sentence.

- We propose a MMT-VQA multi-task learning framework to explicitly model the probing signal in MMT to strengthen the interactions between text and visual modalities.

- Experimental results on Multi30K En-De and En-Fr tasks demonstrate the effectiveness both in terms of BLEU&METEOR scores and specific testsets.

## 2 An Empirical Study on Advanced Vision Features

The use of stronger vision models as image encoders in MMT has gained widespread attention. Li et al. (2022a) demonstrated that the application of more robust pre-trained vision models, such as the Vision Transformer (ViT) (Dosovitskiy et al., 2021), considerably enhances MMT systems, as evidenced by the substantial improvements in the accuracy metrics of the proposed probing tasks. Continuing along this research trajectory, we question whether more recent pre-trained vision models can offer additional advantages to MMT systems, a topic yet to be fully explored. In this context, we chose MAE (He et al., 2022) due to its notable impact in the field of computer vision, as well as two vision-language pre-trained models, namely CLIP (Radford et al., 2021) and BLIP (Li et al., 2022b), for their potential as vision encoders in a MMT system.

Table 1 shows comparisons on the Test2016 test set of the Multi30K En-De task, based on the selective attention model (Li et al., 2022a). We see that the MAE-based MMT system delivers the best performance. This outcome was unexpected as we anticipated models like CLIP, endowed with strong cross-modal modeling capabilities, to excel. However, as shown in Table 2, in probing tasks with

| Feature | Test2016 | Test2017 | MSCOCO |
|---|---|---|---|
| ViT-base | 40.68 | 33.35 | 28.61 |
| MAE-base | **41.65** | **34.27** | **29.75** |
| BLIP-base | 41.01 | 33.73 | 29.41 |
| CLIP-base | 41.28 | 33.83 | 29.66 |

Table 1: BLEU scores of Multi30K En-De task with Selective Attention applied.

incomplete text, CLIP showcased a significant performance boost due to its innate cross-modal modeling knowledge. Therefore, we can find that the cross-modal knowledge of the pre-trained model does not seem to work when the text is complete. This leads us to a fundamental question: *So now that we have better image representation, how can we further enhance the cross-modal interaction ability of the model within the MMT context?*

## 3 MMT-VQA Multi-Task Learning Framework

### 3.1 Probing Signal Modeling: Probing to VQA

Indeed, a compelling proposition is to compel the MMT system to consider image representation even in the absence of ambiguity. The Probing Task proposed by Li et al. (2022a) serves as an effective evaluation metric, determining the usefulness of image features in a limited textual context. Figure 1 shows how the probing task implicitly guides the model to fill the masked position: "probing" the image information. Li et al. (2022a) demonstrated that the probing task efficiently modeling a cross-modal interaction scenario. An intriguing idea is, is it possible to use this task for training?

Probing task can spur and assess cross-modal interaction capabilities of a model simultaneously, making it an effective method for capability enhancement. However, in small datasets like Multi30K, the context information reduction and text coherence decrease during supervised training can significantly impact performance. Therefore, we should transform the Probing Task to other form. So how to extract the probing signal for training is necessary to explore.

We propose a model for the the explicit of probing signals. Figure 1 shows a case of VQA data. The way to model is to formulate questions based on the source, ensuring that all question content originates from the source, and the answer is the masked word in probing task (such as "yellow"

| Model Type | Feature | Test2016 | | | Test2017 | | | MSCOCO | | |
|---|---|---|---|---|---|---|---|---|---|---|
| | | Restrict | Relaxed | BLEU | Restrict | Relaxed | BLEU | Restrict | Relaxed | BLEU |
| **Color-based** | | | | | | | | | | |
| Vision PTM | ViT-base | 42.70 | 52.51 | 38.29 | 29.92 | 45.67 | 31.27 | 32.81 | 42.19 | 28.00 |
| | MAE-base | 49.46 | 61.00 | 39.75 | 34.12 | 53.54 | 32.69 | 37.50 | 56.25 | 29.02 |
| VL-PTM | BLIP-base | 28.54 | 36.17 | 38.03 | 21.52 | 35.43 | 31.03 | 18.75 | 18.75 | 28.83 |
| | CLIP-base | 54.68 | 68.41 | 40.02 | 37.80 | 58.27 | 33.57 | 40.62 | 56.25 | 28.48 |
| **Character-based** | | | | | | | | | | |
| Vision PTM | ViT-base | 68.76 | 74.18 | 38.50 | 64.02 | 67.74 | 30.96 | 67.19 | 71.09 | 27.87 |
| | MAE-base | 63.77 | 69.33 | 38.58 | 62.03 | 67.00 | 31.86 | 65.62 | 69.53 | 28.07 |
| VL-PTM | BLIP-base | 59.20 | 64.05 | 38.06 | 54.84 | 59.55 | 31.69 | 60.55 | 65.62 | 27.80 |
| | CLIP-base | 74.18 | 80.31 | 39.84 | 75.93 | 80.15 | 32.54 | 82.03 | 85.94 | 28.24 |

Table 2: The BLEU scores and the accuracy of Multi30K En-De task of Probing Tasks proposed by Li et al. (2022a) with Selective Attention applied.

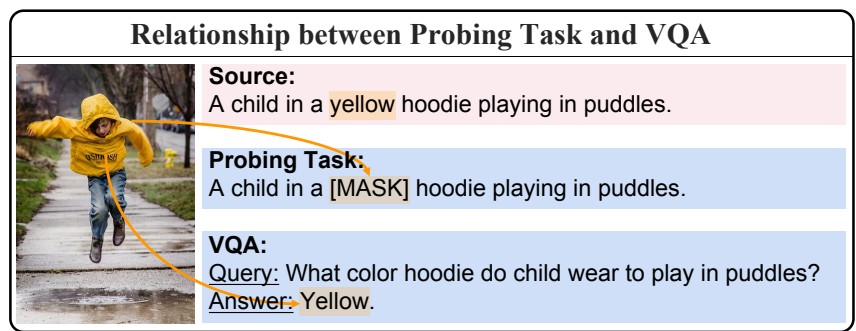

Figure 1: An illustration of the probing task and the VQA task given the source-image pair.

in Figure 1). We thus constructed a VQA-style Question-Answering pair, transforming the original source (which required masking) into a parallel QA pair. Finally, we completed the explicit extraction and representation of the probing signal.

### 3.2 Multi30K-VQA: LLM-Driven Data Generation

To explicitly strengthen the correlations with the help of probing signals, we need solve the challenge of how to generate high-quality question-answering pairs. The key point of the issue lies in the manual selection of appropriate tokens for questioning, a task that proved to be both time-consuming and difficult to implement.

Due to the limitations of supervised training with small-scale data, we turned our attention to large language models (LLMs). Recently, there has been a surge in the capabilities of LLMs, such as GPT3.5 (Ouyang et al., 2022). One can design specific prompting to guide the generation results by providing several task-specific instructions and demonstrations. Here, we summary three important aspects for constructing high-quality QA pairs: determine the answer words, comprehending the

source, and posing questions based on the source.

The Figure 2 illustrates the prompt we employ, and we will now explain our design approach step by step. To guide the LLM to achieve this task step by step, we constructed a reading scenario to describe the requirements. With the source of each sample in the dataset acting as the reading comprehension passage, the LLM formulates questions based on the passage and generates answers to construct QA pairs.

**Task Description** This section primarily constructs a reading comprehension scenario and provides a comprehensive description of the problem. We engaged the Language Learning Model (LLM) in the role of an experienced author of professional English test papers, specifically focusing on reading comprehension questions. We then designated four types of probing questions to guide the LLM in ascertaining the correct answer. In the final step, we charged the LLM with the task of generating a reading comprehension question and supplying an answer rooted in the content of the source sentence.

**Generation Requirements** During this phase, our principal focus was the enhancement of

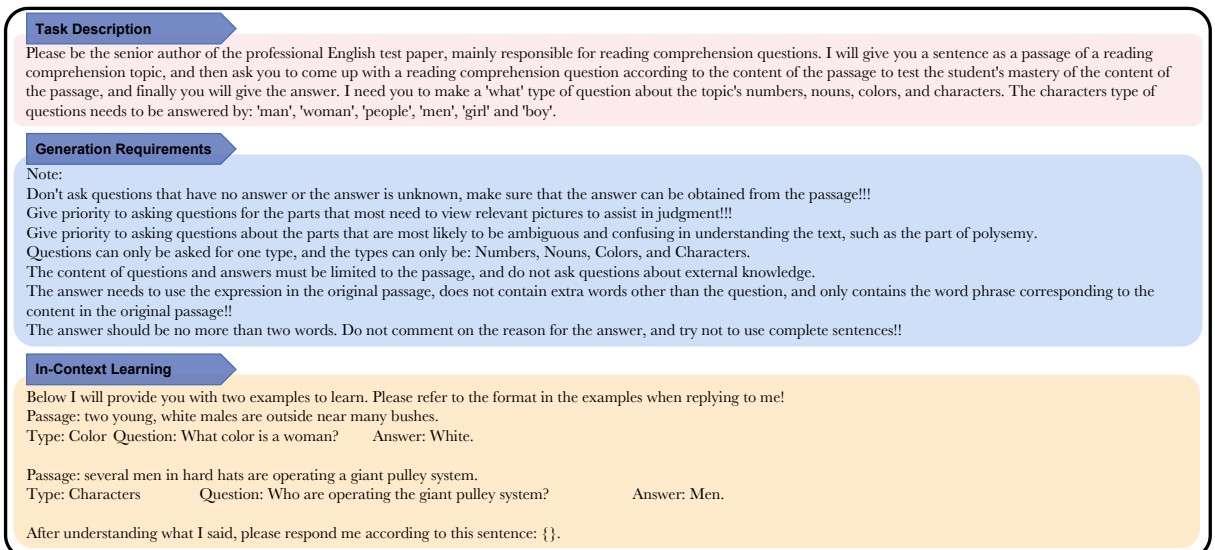

Figure 2: The prompt utilized for generating Question-Answering (QA) pairs in accordance with the provided source sentence using Large Language Models (LLMs). The prompt consists of three parts: task description, generation requirements and the demonstration.

| Source | Type | Question | Answer |
|---|---|---|---|
| A group of fourteen is assembled in a hall with dining tables and a stage. | Number | How many people are in the group? | Fourteen. |
| Two young , white males are outside near many bushes. | Noun | What are the males near? | Bushes. |
| A little girl climbing into a wooden playhouse. | Character | Who is climbing into the wooden playhouse? | Girl. |
| A man in a blue shirt is standing on a ladder cleaning a window. | Color | What color is the man's shirt? | Blue. |

Table 3: Four types of cases in the Multi30K-VQA dataset.

prompts through human feedback. We implemented a set of generative guidelines designed to augment data quality. Notably, we hand-crafted several strategies (as illustrated in the blue segment of Figure 2) for identifying and addressing translation pitfalls, thereby enhancing the overall quality of the generated content.

**In-Context Learning** In the final phase, we selected two carefully designed examples as the demonstration to foster in-context learning for the Large Language Model (`text-davinci-003`). This process could be regarded as a 2-shot prompting strategy. Consequently, we obtained $29,000$ data entries in a Type-Question-Answer format, where several samples are shown in the Table 3. Note that we set the `temperature` to 0, and utlimately obtained the **Multi30K-VQA** dataset, which could be found in our supplementary.

### 3.3 MMT-VQA Multi-Task Learning Model

After obtaining the Multi30K-VQA dataset, our central design is to enable the MMT model to "ask questions to image" via a multi-task learning framework. This leads us to the main task: the con-struction of a model for MMT, while VQA serves as a secondary, supporting task. Upon alignment of data from both tasks, we work with three inputs: `Source`, `Question` and `Image`. While, the corresponding target labels for MMT and VQA are `Target` and `Answer`.

We adopted the selective attention of Li et al. (2022a)'s work as its strong performance. In this way, our model leverages a Transformer encoder in combination with an advanced vision model as the encoder for text and visual modalities respectively. By sharing the parameters of the Text Encoder, we enhance the ability of MMT to facilitate an interplay of information between the two encoders, catering to the needs of VQA for modeling the question.

Given the differences in the fusion methods of the two tasks' information and the variances in the output languages of the tasks, we acknowledged potential significant effects on the performance of model due to discrepancies between the two language representation spaces. As a consequence, we independently initialize the parameters in Selective Attention layer and Decoder.

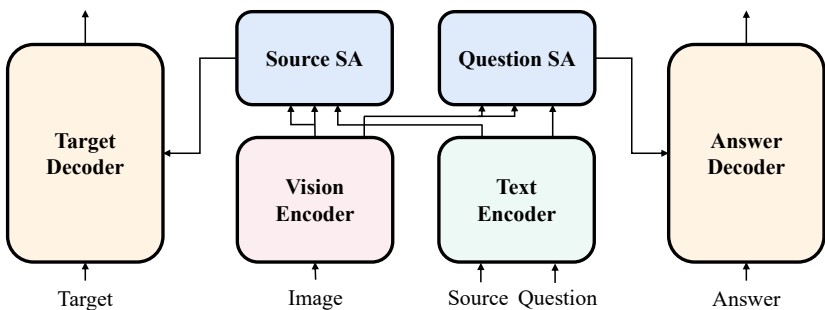

Figure 3: The overall architecture of MMT-VQA multi-task learning framework. SA indicates the Selective Attention proposed by Li et al. (2022a).

**Model Architecture** The proposed overall model architecture is depicted in Figure 3. The training process is initiated by feeding two distinct textual inputs, namely Source ($X^{\text{src}}$) and Question ($X^{\text{qsn}}$), into the Text Encoder. This operation results in two corresponding text representations, $H^{\text{src}}$ and $H^{\text{qsn}}$. Simultaneously, we apply a pre-trained vision model to the associated image, thereby obtaining $H^{\text{img}}$.

$$H^{\text{src}} = \text{TextEncoder}(X^{\text{src}}), \quad (1)$$
$$H^{\text{qsn}} = \text{TextEncoder}(X^{\text{qsn}}), \quad (2)$$
$$H^{\text{img}} = W \cdot \text{PTM}(X^{\text{img}}), \quad (3)$$

where W is a projection matrix to convert the shape of $\text{PTM}(X^{\text{img}})$ into that of $X^{\text{qsn}}$ and $X^{\text{src}}$. $\text{PTM}(\cdot)$ denotes MAE or other advanced vision models.

After obtaining the three representations, the text is associated with the image patch using a single-head attention network, where the queries derive from $H^{\text{src}}$ and $H^{\text{qsn}}$, respectively, and the key and value are both $H^{\text{img}}$. Formally, these two processes can be expressed as:

$$H^{\text{img}}_{\text{attn}} = \text{Softmax}(\frac{QK^{\text{T}}}{\sqrt{d_k}})V, \quad (4)$$

where $d^{\text{k}}$ is the same as the dimension of $X^{\text{src}}$ or $X^{\text{qsn}}$ because a single head is used.

Then calculate two gate and the fused output as follow:

$$\lambda = \text{Sigmoid}(UH^{\text{text}} + VH^{\text{img}}), \quad (5)$$
$$H^{\text{Out}} = (1 - \lambda) \cdot H^{\text{text}} + \lambda \cdot H^{\text{img}}_{\text{attn}}, \quad (6)$$

where $H^{\text{text}}$ is $H^{\text{src}}$ or $H^{\text{qsn}}$. U and V are trainable variables. $\lambda$ controls the mixing ratio of visual information. Then, the fusion vectors $H^{\text{Out}}_1$ and $H^{\text{Out}}_2$ computed via Eq. 6 are sent to the Target Decoder and the Answer Decoder, to predict the target sentence and answer, respectively.

**Training Objective** In order to facilitate the learning process of MMT with the assistance of VQA, we regard MMT as the primary task and VQA as an auxiliary task. The training objective for MMT is the standard Maximum Likelihood Estimation (MLE) loss, while simultaneously optimizing the training objective for VQA in conjunction with the MMT loss:

$$\mathcal{L}_{\text{Total}} = \mathcal{L}_{\text{MMT}} + \lambda\mathcal{L}_{\text{VQA}}, \quad (7)$$

where $\lambda$ is a hyperparameter controlling the balance between the MMT loss and the VQA loss. During training, the training objective is to minimize the total loss $\mathcal{L}_{\text{Total}}$.

## 4 Experiments

### 4.1 Data

**Multi30K** We evaluate our methods on two standard benchmarks: Multi30K English-German (En-De) and English-French (En-Fr) (Elliott et al., 2016). Multi30K is a widely used MMT dataset, containing $31,014$ images with one English description and the manual translation in German and French. The training and validation sets consist of $29,000$ and $1,014$ instances, respectively. We reported the results on the Test2016, Test2017, Test2018 and MSCOCO test sets (Elliott et al., 2017), which includes $1,000$, $1,000$, 1071 and 461 instances, respectively.

**Multi30K-VQA** As detailed in Section 3.2, we introduce the Multi30K-VQA dataset, featuring $29,000$ rigorously curated QA pairs (the same size as the training set) for text-image tasks. To ensure quality and minimize hallucination, our LLM-generated QA pairs underwent iterative reviews and refinements on 500 randomly selected samples. Despite these efforts, there are even $1,012$ pairs fell short of our expectations. To address this issue,

| # | System | Feature | English->German | | | | | | English->French | | | | | |
|---|--------|---------|------|---|------|---|------|---|------|---|------|---|------|---|
| | | | Test2016 | | Test2017 | | MSCOCO | | Test2016 | | Test2017 | | MSCOCO | |
| | | | B | M | B | M | B | M | B | M | B | M | B | M |
| | Text only | | | | | | | | | | | | | |
| 1 | Transformer Tiny (Vaswani et al., 2017) | - | 41.02 | 68.22 | 33.36 | 62.05 | 29.88 | 56.64 | 61.80 | 81.02 | 53.46 | 75.62 | 44.52 | 69.43 |
| | Existing MMT Systems | | | | | | | | | | | | | |
| 2 | Doubly-ATT (Calixto et al., 2017) | ResNet | 41.45 | 68.04 | 33.95 | 61.83 | 29.63 | 59.21 | 61.99 | 81.12 | 53.72 | 75.71 | 45.16 | 70.25 |
| 3 | Imagination (Elliott and Kádár, 2017) | ResNet | 41.31 | 68.06 | 32.89 | 61.29 | 29.90 | 56.57 | 61.90 | 81.20 | 54.07 | 76.03 | 44.81 | 70.35 |
| 4 | UVR-NMT (Zhang et al., 2020) | ResNet | 40.79 | - | 32.16 | - | 29.02 | - | 61.00 | - | 53.20 | - | 43.71 | - |
| 5 | Gated Fusion (Wu et al., 2021) | ResNet | 41.96 | 67.84 | 33.59 | 61.94 | 29.04 | 56.15 | 61.69 | 80.97 | 54.85 | 76.34 | 44.86 | 70.51 |
| 6 | Selective Attention (Li et al., 2022a) | ViT-base | 41.93 | 68.55 | 33.60 | 61.42 | 31.14 | 56.77 | 62.48 | 81.71 | 54.44 | 76.46 | 44.72 | 71.20 |
| 7 | IKD-MMT (Peng et al., 2022) | ResNet | 41.28 | 58.93 | 33.83 | 53.21 | 30.17 | 48.93 | 62.53 | 77.20 | 54.84 | 71.87 | - | - |
| 8 | PLUVR (Fang and Feng, 2022) | ResNet | 40.30 | - | 33.45 | - | 30.28 | - | 61.31 | - | 53.15 | - | 43.65 | - |
| 9 | IVA (Ji et al., 2022) | ResNet | 41.77 | 68.60 | 34.58 | 62.40 | 30.61 | 56.70 | - | - | - | - | - | - |
| 10 | VALHALLA (Li et al., 2022c) | VQGAN VAE | 42.60 | 69.30 | 35.10 | 62.80 | 30.70 | 57.60 | 63.10 | 81.80 | 56.00 | 77.10 | 46.40 | 71.30 |
| 11 | Noise-robust (Ye et al., 2022) | Resnet | 42.56 | 59.98 | 35.09 | 54.51 | 31.09 | 50.46 | 63.24 | 77.54 | 55.48 | 72.62 | 46.34 | 67.40 |
| 12 | Multimodal-mixup (Ye and Guo, 2022) | Resnet | 41.77 | 58.93 | 33.07 | 51.85 | 29.90 | 49.09 | 62.23 | 76.85 | 55.18 | 73.37 | 44.42 | 66.41 |
| | Our MMT Systems | | | | | | | | | | | | | |
| 13 | Selective Attention | MAE-base | 41.65 | 68.43 | 34.27 | 62.08 | 29.75 | 56.68 | 61.87 | 80.76 | 54.07 | 76.10 | 44.65 | 70.43 |
| 14 | MMT-VQA | MAE-base | 42.55 | 69.00 | 34.58 | 61.99 | 30.96 | 57.23 | 62.24 | 81.77 | 54.89 | 76.53 | 45.75 | 71.21 |

Table 4: BLEU (B) and METEOR (M) scores of Multi30K En-De and En-Fr tasks. Some of the results are from Li et al. (2022a)'s work.

| System | Feature | English->German | | English->French | |
|--------|---------|------|------|------|------|
| | | B | M | B | M |
| Transformer-tiny | - | 30.52 | 57.37 | 38.05 | 64.01 |
| Selective Attention | ViT | 30.41 | 57.29 | 38.10 | 64.17 |
| Selective Attention | MAE | 31.09 | 58.17 | 38.18 | 64.55 |
| MMT-VQA | MAE | 31.74 | 58.49 | 38.72 | 64.87 |

Table 5: BLEU (B) and METEOR (M) scores of Multi30K En-De and En-Fr tasks on the Test2018 test set.

| Type | Noun | Character | Color | Number |
|------|------|-----------|-------|--------|
| Count | 5133 | 18423 | 5303 | 141 |

Table 6: Answer type statistics of Multi30K-VQA.

| Feature | Method | Test2016 | Test2017 | MSCOCO |
|---------|--------|----------|----------|--------|
| ViT-base | MMT | 40.68 | 32.49 | 28.61 |
| | MMT-VQA | $41.14^{\uparrow 0.46}$ | $33.35^{\uparrow 0.86}$ | $28.53^{\downarrow 0.08}$ |
| MAE-base | MMT | 41.65 | 34.27 | 29.75 |
| | MMT-VQA | $42.55^{\uparrow 0.90}$ | $34.58^{\uparrow 0.31}$ | $30.96^{\uparrow 1.21}$ |
| BLIP-base | MMT | 41.01 | 33.73 | 29.41 |
| | MMT-VQA | $41.91^{\uparrow 0.90}$ | $34.03^{\uparrow 0.30}$ | $29.81^{\uparrow 0.40}$ |
| CLIP-base | MMT | 41.28 | 33.83 | 29.66 |
| | MMT-VQA | $41.43^{\uparrow 0.15}$ | $33.90^{\uparrow 0.07}$ | $30.32^{\uparrow 0.66}$ |

Table 7: BLEU scores of Multi30K En-De task of MMT-VQA model compared with MMT (Selective Attention) model with Different Pre-trained Vision Models.

we employed hand-crafted rules to refine 605 of these pairs and annotated the remaining 407 with sentence-level annotations via 3 annotators. Our final prompt, fine-tuned through 5 iterations, yielded the best output among various tests. Answer-type distributions are shown in Table 6.

## 4.2 Model Settings

Considering the small size of the Multi30K data set, we follow the prior work using Transformer-Tiny as the base configuration (Wu et al., 2021). In general, the MMT-VQA Multi-task Learning Model consists of a Pre-Trained Image Encoder, a 4-layer Text Encoder, two Selective Attention layers, and two 4-layer Decoders. For each encoder and decoder layer, the hidden size is set to 128, the intermediate size of Feed-Forward Networks is set to 256, and the number of heads is set to 4.

Our implementation was based on Fairseq library (Ott et al., 2019), supplemented by the integration of Vision Pre-trained Models from the Huggingface[1]. The optimization process is facilitated by the Adam optimizer, with parameters $\beta_1$ and $\beta_2$ set at 0.9 and 0.98 respectively, and $\epsilon$ established at $10^{-8}$. Learning rate is set to 0.005 and a warmup phase consisting of 2000 update steps. The model incorporates a dropout rate of 0.3 and label-smoothing of 0.1. The training process is conducted on 2 GeForce RTX 3090 GPUs, with each training batch comprising 4096 tokens per GPU. To ensure a fair comparison with the previous work, we adopt an early stopping strategy, triggered if there is no performance improvement on the validation set over 10 epochs. For inference, we average the last 10 checkpoints for robust performance. Note that our $\lambda$ is set to 0.3, which achieves the best in our preliminary experiments.

[1]https://huggingface.co/

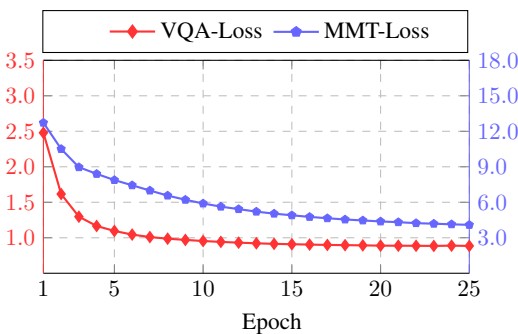

Figure 4: The change curve of two loss in the training process of multi-task learning model.

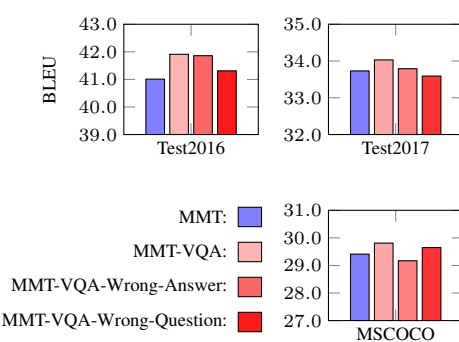

Figure 5: BLEU scores of different data processes to *Question* and *Answer*.

## 4.3 Results

Table 4 presents the results of various MMT systems evaluated on the Multi30K dataset. All models were evaluated in terms of BLEU and METEOR scores on six separate test sets for En-De and En-Fr tasks. Upon comparison with the most recent and strong prior works, MMT-VQA can outperform most of competing MMT systems in terms of the BLEU and METEOR metrics, demonstrating the effectiveness of strengthening cross modality interactions. For a more convincing result, we further conducted experiments on Test2018 in Table 5, where similar phenomenon could be observed.

Experimental results in Table 7 demonstrate the general applicability of MMT-VQA multitask learning framework. We can see that MMT-VQA is often beneficial when transitioning to other advanced vision features, encompassing both pretrained vision models and vision-language pretrained models, such as CLIP and BLIP. The MMT-VQA method exhibits improvements across various test sets, with the magnitude of these enhancements varying. Remarkably, the most substantial increment observed was as high as 1.21, thus reinforcing the general applicability and robustness of our proposed methodology.

## 5 Analysis

Here, we aim to take more in-depth analysis on the importance of MMT-VQA. Without specific notion, all results were conducted on Multi30K En-De task.

### 5.1 Illustrating Training Loss

First, we would like to validate whether the improvements come from the strengthened interaction between modalities or just act as an regularization signal. Figure 4 plots the curves of two training

losses, $\mathcal{L}_{\text{MMT}}$ (in blue) and $\mathcal{L}_{\text{VQA}}$ (in red) during the training. Given the minimal changes observed in the latter stages of training, we have chosen to present only the trends from the initial 25 epochs. It is evident that both models were able to converge stably at a certain level, validating that the auxiliary loss is not a regularization. We also observed a rapid convergence during the training of the VQA tasks. This can be attributed to our strategic choice of using shorter vocabulary as answer words during data generation, which significantly reduced the learning difficulty for the model.

### 5.2 Incongruent Question-Answering Pairs

We further explored the effects of incongruent question-answering pairs training to substantiate our claims. Under the assumption that if VQA acts as a noise signal during training, randomizing the correct Question and Answer in the training data should not significantly impact the results. To scrutinize this, we conducted additional experiments, employing the BLIP model for visual feature extraction due to its notable performance improvement. The experimental results, depicted in Figure 5, reveal a significant impact on the performance of MMT-VQA-Wrong-Answer and MMT-VQA-Wrong-Question across all three test sets. Some metrics even regressed below the performance of the Selective Attention, which did not employ the MMT-VQA method. We attribute these performance decreases to the failure of model to correctly learn the correspondences and probing signals among the Question, Answer and Image triad. This further emphasizes the importance of correct alignment and interaction in improving the efficacy of MMT systems.

| Model | Process | Test2016 | Test2017 | MSCOCO |
|---|---|---|---|---|
| MMT | - | 41.01 | 33.73 | 29.41 |
| MMT-VQA | - | 41.91 | 34.03 | 29.81 |
| MMT-VQA | Question->Source | 41.21 | 32.74 | 29.07 |

Table 8: BLEU scores of exploring whether the improvement comes from the introduction of Questions in the training phase and contains other information.

## 5.3 Is VQA really helping for cross-modal interaction?

Since the complete VQA task input consists of two parts of data, Question and Image, we intend to evaluate whether VQA brings stronger cross-modal interaction capabilities to the MMT model from the perspective of processing the two data.

**Question Perspective** A defining feature of MMT-VQA compared to traditional MMT models is the added question input, thus raising the concern: *Is MMT-VQA's performance boost due to this extra information?* We validated this by replacing all questions in the Multi30K-VQA dataset with source text during training. Table 8 (row 3) shows the model's En-De task performance, with the first two rows for comparison against baseline MMT and MMT-VQA. The substitution leads to a marked decline in performance across all test sets. Against the baseline, minor gains appeared on only one test set, while losses were significant on the others. These results confirm that MMT-VQA's improvements are not solely due to the added question data, but rather indicate the model's ability to leverage semantic cues from questions for enhanced multimodal interactions.

**Image Perspective** We evaluated the visual context sensitivity using adversarial methods, following the inconsistent decoding setup in prior studies (Caglayan et al., 2019, 2021). By replacing random proportions of image samples with irrelevant images, we performed ablation experiments and compared BLEU performance changes between the MMT baseline (Selective Attention) and MMT-VQA methods. As seen in Figure 6, without our strengthened modality interactions, the baseline shows higher sensitivity and more overall fluctuation without a clear downward trend, suggesting inadequate use of image information. On the other hand, MMT-VQA, despite similar fluctuation levels, maintained a downward trend, indicating stronger dependence on visual information accuracy and improved visual modality interaction.

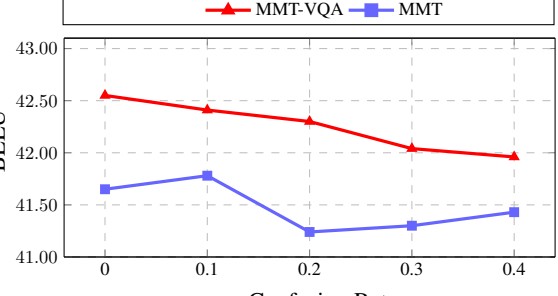

Figure 6: BLEU scores of the two models on the Test2016 test set with different levels of confusion.

| # | System | Feature | Test2016-con |
|---|---|---|---|
| 1 | Transformer Tiny | - | 37.89 |
| 2 | MMT | BLIP-base | 38.53 |
| 3 | MMT-VQA | BLIP-base | $39.62^{\uparrow 1.09}$ |
| 4 | MMT | MAE-base | 39.12 |
| 5 | MMT-VQA | MAE-base | $\mathbf{39.71}^{\uparrow 0.59}$ |

Table 9: BLEU scores of Multi30K En-De task on the Test2016-con test set.

## 5.4 Test2016-con: LLM-Driven Model Evaluation

We used LLMs to identify confusing words within samples of Test2016 and subsequently formed a subset, Test2016-con, which consists of samples with these words. This subset prioritizes image disambiguation, necessitating the interplay of two modalities. Table 9 illustrates that our MMT-VQA method notably exceeds the performance of the baseline system with MAE and BLIP vision features, suggesting a stronger integration with image modality for disambiguation. It is noteworthy that we will also open-source this testset in our codebase.

## 5.5 Case Study

We also make a quantitative case study as shown in the Table 10. In the first case, we observed that the MMT system, despite employing a selective attention mechanism and Vision Transformer (ViT) features, faltered in the translation of the word "up" in the German context. Specifically, it omitted this word entirely. In contrast, our MMT-VQA system produced the accurate German translation "hochgelegten", which effectively captures the notion of "elevated" or "up". Another noteworthy observation is that conventional MMT systems struggle with accurate color translation when a sentence contains multiple color descriptors. This limitation was effectively mitigated in our MMT-VQA sys-

| | | |
|---|---|---|
| 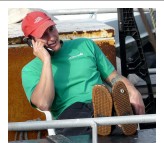 | SRC | : A man talks on the phone with his feet up . |
| | REF | : Ein mann telefoniert mit hochgelegten füßen . |
| | `MMT(SA)` | : Ein mann telefoniert mit ~~den~~ füßen . |
| | `MMT-VQA` | : Ein mann telefoniert mit **hochgelegten** füßen . |
| 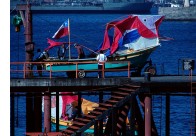 | SRC | : A boat with red, white and blue sails docking at a pier . |
| | REF | : Ein boot mit roten , weißen und blauen segeln dockt an einem pier an . |
| | `MMT(SA)` | : Ein boot mit ~~rot blauen~~ segeln an einem pier . |
| | `MMT-VQA` | : Ein boot mit **roten , weißen und blauen** segeln an einem pier . |

Table 10: Qualitative examples to demonstrate enhanced cross-modal capability. ~~Strikethrough~~ and **bold** words present the incorrect and good lexical choices.

tem by the incorporation of probing signals. These signals, particularly those covering color question-answering pairs, significantly enhanced the system's ability to resolve such translation ambiguities. Our analysis suggests that the failure of the MMT system with selective attention in translating seemingly simple sentences stems primarily from its inability to disambiguate certain words effectively.

## 6 Related Work

The task of MMT forms an intersection between NLP and CV, with the goal of enhancing translation outcomes through the integration of image-based context. The earlier approaches focused primarily on the effective amalgamation of these dual-modal representations (Calixto and Liu, 2017; Elliott and Kádár, 2017; Delbrouck and Dupont, 2017). Subsequent research, however, began to question the significance of visual modality, finding that irrelevant images did not substantially influence translation quality. Indeed, the absence of images did not lead to a substantial drop in BLEU scores (Elliott, 2018). Caglayan et al. (2019) provided further insight into this phenomenon by demonstrating that visual modality could offer utility in contexts with limited linguistic data, but its influence was less pronounced when full sentences were available.

In recent studies, Wu et al. (2021) attributed improvements to the effects of regularization during training. They further showed that even a tiny configuration could outperform earlier models in terms of BLEU score. Concurrently, additional efforts include the use of hallucination representations augmentation (Li et al., 2022c), Noise-Input to mask the additional noise thereby guiding the model to focus on meaningful interactions (Ye et al., 2022), and Dual-level interactive mixup for feature alignment (Ye and Guo, 2022) have emerged. However, existing MMT systems still face difficulties when dealing with complete textual contexts, a concern that also forms the crux of the present study. Distinctively, we focus on fortifying text-visual interactions through the deployment of probing signals during training. Our approach shares thematic similarity with Ji et al. (2022), who explored this issue from a mutual information standpoint.

## 7 Conclusions

In this work, we offer a novel perspective on multimodal machine translation (MMT), attributing reduced system sensitivity to visual information in complete text inputs to insufficient cross-modal interaction rather than image information redundancy. We propose a unique approach that generates Visual Question-Answering (VQA) style pairs from source text via LLMs to model the probing signal in MMT explicitly. As a bonus, we propose a Multi30K-VQA dataset and design a MMT-VQA multitask learning framework. Our methodology is validated by experimental results on two representative benchmarks, evidencing its effectiveness and driving forward the MMT research.

## Acknowledgments

This work was supported in part by the National Science Foundation of China (No.62276056), the National Key R&D Program of China, the China HTRD Center Project (No.2020AAA0107904), the Natural Science Foundation of Liaoning Province of China (2022-KF-16-01), the Yunnan Provincial Major Science and Technology Special Plan Projects (No.202103AA080015), the Fundamental Research Funds for the Central Universities (Nos. N2216016, N2216001, and N2216002), and the Program of Introducing Talents of Discipline to Universities, Plan 111 (No.B16009).

## Limitations

The performance of our proposed model will depend on the quality of the generated dataset, and the quality of the data set mainly depends on the hints we prepare for the large model. Therefore, although we make good use of the ability of large model, how to design a better prompt and reduce the influence of human factors on prompt quality is still worth exploring in the future. In addition, we haven't explored a better model architecture in depth, and it is very likely that the model we proposed is not the most suitable for this task.

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

## A Sample of Multi30K-VQA

Table 11 shows samples that we have picked out from the Multi30K-VQA dataset.

## B Case Study on Type of Error

We try to figure out what type of error additional VQA training helps the most. We conduct case studies from three perspective:

**Perspective of Part-of-Speech (POS):** We analyzed translation quality using POS tagging. For each POS type in the hypothesis, we counted its occurrence in sentences and then segmented the test set accordingly. We compared BLEU scores for the MMT-VQA and MMT (Selective Attention) methods across these subsets. To isolate the effect of specific POS types, we calculated the score differences and adjusted for word frequency. Our results highlight that adjectives (JJ) and plural nouns (NNS) most significantly improve translation quality when using the VQA method.

**Perspective of Long and Difficult Words:** We found that $3\%$ of vocabulary consists of long and difficult words. After calculating that the 97th percentile of all word lengths is 9, we divided the test set based on whether sentences contained long and difficult words. We found that the MMT-VQA method improved the BLEU score by $0.98$ points on the test subset with long and difficult words compared to the selective attention baseline method (Li et al., 2022a). The improvement on the subset without long and difficult words was $0.67$ points.

**Perspective of Long and Difficult Sentences:** We then completed the identification of long and difficult sentences. We first determined the number of clusters for sentence length categorization to be 2 based on the elbow method, dividing sentence lengths into two categories: long sentences and short sentences. We found that during the translation of long sentences, the BLEU score difference between the VQA and SA methods was $1.08$, and for short sentences, it was $0.63$—nearly a $70\%$ improvement.

| | |
|---|---|
| 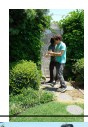 | Source   : Two young , white males are outside near many bushes .
Question: What are the males near?
Answer   : Bushes.
Type     : Noun. |
| 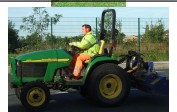 | Source   : A man in a neon green and orange uniform is driving on a green tractor .
Question: What colors are on the man's uniform?
Answer   : Neon green and orange.
Type     : Color. |
| 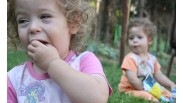 | Source   : Two young toddlers outside on the grass .
Question: Who is outside on the grass?
Answer   : Toddlers.
Type     : Character. |
| 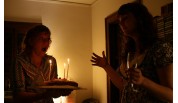 | Source   : A young man is presenting another woman with a cake with three candles on top .
Question: How many candles are on the cake?
Answer   : Three.
Type     : Number. |
| 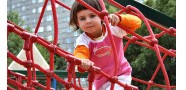 | Source   : The small child climbs on a red ropes on a playground .
Question: What is the child climbing on?
Answer   : Rope.
Type     : None. |
| 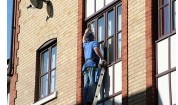 | Source   : A man in a blue shirt is standing on a ladder cleaning a window .
Question: What color is the man's shirt?
Answer   : Blue.
Type     : Color. |
| 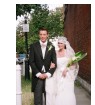 | Source   : A beautiful bride walking on a sidewalk with her new husband .
Question: Who is walking on the sidewalk?
Answer   : Bride and husband.
Type     : Character. |
| 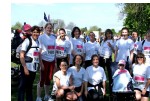 | Source   : A sports team made up of 14 women waring white t-shirt and pink tags .
Question: How many women are on the sports team?
Answer   : 14.
Type     : Number. |
| 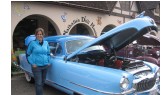 | Source   : Woman in blue jacket shows off her vintage car in front of nathalie's dollhouse .
Question: What is the woman showing off?
Answer   : Vintage car.
Type     : Noun. |
| 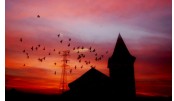 | Source   : A beautiful photograph of a church at sunset with birds flying overhead and orange-red sky .
Question: What color is the sky?
Answer   : Orange-red.
Type     : Color. |
| 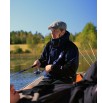 | Source   : A fisherman is reeling his rod while another relaxes in a boat on water .
Question: Who is reeling the rod?
Answer   : Fisherman.
Type     : Character. |
| 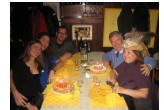 | Source   : A group of people at a party with two cakes on the table .
Question: How many cakes are on the table?
Answer   : Two.
Type     : Number. |

Table 11: Qualitative samples of Multi30K-VQA.