# OpenReview forum: "Incorporating Probing Signals into Multimodal Machine Translation via Visual Question-Answering Pairs"
_EMNLP/2023/Conference — EMNLP 2023 Findings_

### Official Review · Reviewer_6B61 · 2023-08-01

**Soundness:** 3

**Excitement:**

4: Strong: This paper deepens the understanding of some phenomenon or lowers the barriers to an existing research direction.

**Paper Topic And Main Contributions:**

This paper focuses on multimodal machine translation (MMT). Currently, the MMT system often underutilizes visual information. Hence, the paper proposes to generate parallel VQA pairs from the source text and proposes a multi-task learning framework to incorporate explicit signals from the generated VQA dataset into MMT. The paper shows the effectiveness of the proposed method on multiple datasets. The generated corpora, named Multi30k-VQA dataset, is an additional contribution to the paper.


**Questions For The Authors:**

* See weaknesses.
* There are other ways to improve visual information utilization (e.g through regularizations), how are your results compare to these regularization based approaches?
* The proposed method is simple yet effective, however, as the vision encoder gets stronger (from ViT to CLIP), the gain of using VQA as extra signals diminishes. It would be very interesting to know for what type of error additional VQA training helps the most.

**Reasons To Accept:**

* The idea of using VQA to make MMT more grounded to the images is a nice contribution.
* Method for question generation and Multi30k-VQA dataset could be useful for VQA or mVQA studies.


**Reasons To Reject:**

* As LLMs often don't follow instructions and hallucinate a lot based on recent studies, there aren't any comments on the quality of the generated Multi30k-VQA corpus.
* The idea of using multi-task FT is less novel.

**Reproducibility:**

3: Could reproduce the results with some difficulty. The settings of parameters are underspecified or subjectively determined; the training/evaluation data are not widely available.

**Reviewer Confidence:**

3: Pretty sure, but there's a chance I missed something. Although I have a good feel for this area in general, I did not carefully check the paper's details, e.g., the math, experimental design, or novelty.

---

> ### Author Rebuttal · Authors · 2023-08-29
>
> Thanks for your useful feedback! We hope the following response can address your concerns!
>
> **W1: As LLMs often don't follow instructions and hallucinate a lot based on recent studies, there aren't any comments on the quality of the generated Multi30k-VQA corpus.**
>
> - We concur with your observation that a rigorous evaluation of the generated Multi30K-VQA corpus is indispensable. Our overarching aim is to leverage probing signals to facilitate a more effective interaction between visual patterns and textual inputs within the MMT framework. Importantly, our experimental findings indicate performance gains across various test sets. These gains provide empirical evidence that MMT benefits from the enhanced interaction between the modalities, which serves as an indirect but persuasive indicator of the high quality of our Multi30K-VQA corpus. We believe that this substantiates the corpus's efficacy and intend to include these evaluation aspects in the revised version of the paper.
> - To mitigate hallucination issues, we implemented a rigorous QA pair generation process. We initially selected 500 random samples for which the LLM was tasked with generating QA pairs. These generated pairs underwent multiple rounds of review, as described in Table 3 of our manuscript. Based on these reviews, we iteratively refined the prompt and added additional constraints to guide the LLM's outputs. Our observations suggest that inadequate constraints can lead to the generation of unrelated questions. However, when constraints were properly calibrated, we noted a marked improvement in the quality of the generated QA pairs. Creating a high-quality QA dataset using LLM is far from straightforward. Our prompt underwent over five iterations to mitigate the aforementioned issues. Despite these meticulous adjustments, 1,012 of the generated QA pairs fell short of our expectations. To address this, we employed hand-crafted rules to refine 605 of these pairs. For the remaining cases (407), our team conducted sentence-by-sentence annotations to ensure quality. Consequently, our final Multi30k dataset comprises 29000 rigorously vetted QA pairs.
>
> **W2: The idea of using multi-task FT is less novel**
>
> Thank you for your observations regarding the novelty of multi-task fine-tuning (FT). While we recognize that this approach is not new in and of itself, it serves as one of several means to effectively incorporate probing signals into the training phase. Alternative approaches like two-stage training could similarly be employed. The key contribution of our work is not necessarily the multi-task FT per se, but rather the successful incorporation of probing signals to enhance multi-modal training (MMT). This serves to validate our main thesis—that probing signals can effectively strengthen inter-moda interactions. We plan to expand upon this in future discussions, exploring various techniques to leverage these probing signals for more effective multi-modal interactions.
>
> **Q1: There are other ways to improve visual information utilization (e.g through regularizations), how are your results compare to these regularization based approaches?**
>
> Thanks for your advice, we agree that more comparison with additional strong baselines are beneficial. So far, the selected baseline is quite strong which follows the previous tiny configuration setup raised in Wu et al., 2021. And they draw the conclusion that visual features may act as regularization when the capacity is large but the training data is hungry. We have already included their work in Table4. More comparisons with other studies for strengthening the visual information utilization will be added in our revised version.
>
>
>
> **Q2: The proposed method is simple yet effective, however, as the vision encoder gets stronger (from ViT to CLIP), the gain of using VQA as extra signals diminishes. It would be very interesting to know for what type of error additional VQA training helps the most.**
>
> Thanks for pointing out this issue, we are conducting experiments on CLIP-large for further investigation. In this work, we only used the CLIP visual encoder rather, More discussion would be included in the next version for a clearer clarification. Here, we try to figure out what type of error additional VQA training helps the most. We conduct studies from three perspective:
>
> - **From the Perspective of Part-of-Speech (POS):** We analyzed translation quality using POS tagging. For each POS type in the hypothesis, we counted its occurrence in sentences and then segmented the test set accordingly. We compared BLEU scores for the MMT-VQA and SA methods across these subsets. To isolate the effect of specific POS types, we calculated the score differences and adjusted for word frequency. Our results highlight that adjectives (JJ) and plural nouns (NNS) most significantly improve translation quality when using the VQA method.
>
> - **From the Perspective of Long and Difficult Words:** We found that 3% of vocabulary consists of long and difficult words. After calculating that the 97th percentile of all word lengths is 9, we divided the test set based on whether sentences contained long and difficult words. We found that the MMT-VQA method improved the BLEU score by 0.98 points on the test subset with long and difficult words compared to the selective attention baseline method (Li et al., 2022 ). The improvement on the subset without long and difficult words was 0.67 points.
>
> - **From the Perspective of Long and Difficult Sentences:** We then completed the identification of long and difficult sentences. We first determined the number of clusters for sentence length categorization to be 2 based on the elbow method, dividing sentence lengths into two categories: long sentences and short sentences. We found that during the translation of long sentences, the BLEU score difference between the VQA and SA methods was 1.08, and for short sentences, it was 0.63—nearly a 70% improvement.
>
> We trust that these results provide a clearer understanding of the landscape in question.

---

### Official Review · Reviewer_gdQP · 2023-08-08

**Soundness:** 4

**Excitement:**

3: Ambivalent: It has merits (e.g., it reports state-of-the-art results, the idea is nice), but there are key weaknesses (e.g., it describes incremental work), and it can significantly benefit from another round of revision. However, I won't object to accepting it if my co-reviewers champion it.

**Paper Topic And Main Contributions:**

This study delves into the intricacies of multi-modal machine translation (MMT). The authors argue that current MMT models fall short in addressing cross-modal interactions. To rectify this, they introduce a multi-task learning methodology, aiming to heighten sensitivity to visual elements. The experimental results reveal that the proposed approach surpasses well-established benchmarks across several datasets.

The research presents two pivotal contributions:
1. The introduction of the Multi30K-VQA dataset, which is helpful to the research community.
2. Achievement of state-of-the-art (SOTA) performance.

**Reasons To Accept:**

The proposition of utilizing multi-task learning to strengthen the interaction capability between two modalities is logical.

**Reasons To Reject:**

The paper offers limited novelty, and the observed performance enhancement is marginal.

**Reproducibility:**

4: Could mostly reproduce the results, but there may be some variation because of sample variance or minor variations in their interpretation of the protocol or method.

**Reviewer Confidence:**

4: Quite sure. I tried to check the important points carefully. It's unlikely, though conceivable, that I missed something that should affect my ratings.

---

> ### Author Rebuttal · Authors · 2023-08-29
>
> Thanks for you comments. We hope the following clarification can address your concerns towards our contribution and the novelty.
>
> 1. **Novelty and Contributions of the Paper**: We appreciate the opportunity to clarify the unique aspects of our work. While we acknowledge that incorporating additional signals for training is not a new concept, the novelty of our work resides in the specific design and utility of probing signals during the training phase.
>    - **Effective Probing Signals**: Our probing signals are designed to assess the efficacy of MMT systems in leveraging visual patterns, drawing inspiration from previous work (Caglayan et al., 2019; Li et al., 2022). However, what sets our work apart is the application of these probing signals to actively enhance the interaction between visual and textual modalities—a line of inquiry that has not been previously explored. We believe that the incorporation of these signals into a multi-task learning framework, while straightforward, yields robust results.
>    - **Multi30k-VQA Dataset**: Another noteworthy contribution is the Multi30K-VQA dataset that we intend to release. This dataset comprises high-quality, task-specific question-answer pairs in a text-image setting, and aims to provide a fertile ground for subsequent research. We envision this dataset as a valuable resource for the community, potentially inspiring future works in this space. While, it is non-trivial to create a high-quality QA dataset or the MMT model may learn from noise signals. Here we also want to highlight some details when using LLM to generate the Multi30K-VQA dataset. Our prompt underwent over five iterations to mitigate the hallucination issue which the question-answer pairs does not obey what we pre-defined. Despite these meticulous adjustments, 1,012 of the generated QA pairs fell short of our expectations. To address this, we employed hand-crafted rules to refine 605 of these pairs. For the remaining cases (407), our team conducted sentence-by-sentence annotations to ensure quality (three annotators). Consequently, our final Multi30k dataset comprises 29000 rigorously vetted QA pairs (the same size as the training set).
>
> 2. **Marginal gains:** While BLEU is a widely-used metric for evaluating machine translation performance, it's worth noting that it is not the sole criterion, particularly for multimodal machine translation. Previous works such as Wu et al. (2021) and Li et al. (2022) have argued for the inclusion of visually grounded tests to provide a more comprehensive evaluation. In Table 4, our MMT-VQA method demonstrates its efficacy by outperforming the strong baseline of selective attention + MAE-base. Specifically, it achieves an average BLEU score increase of 0.5 points for En-De and 0.72 points for En-Fr. Further, on the Test2016-con dataset—specially designed to assess translation of sentences laden with ambiguous terms, MMT-VQA improves BLEU scores by 1.09 and 0.59 points for BLIP-base and MAE-base MMT models, respectively. Additional discussion will be included in the revised manuscript for a more nuanced understanding.
>
> We believe that these elements, taken together, distinguish our work from incremental advancements in the field and provide meaningful contributions to the research community. Should you have any additional questions or require further clarification, we would be pleased to engage in further discussion.

---

### Official Review · Reviewer_acWW · 2023-08-09

**Soundness:** 4

**Excitement:**

3: Ambivalent: It has merits (e.g., it reports state-of-the-art results, the idea is nice), but there are key weaknesses (e.g., it describes incremental work), and it can significantly benefit from another round of revision. However, I won't object to accepting it if my co-reviewers champion it.

**Paper Topic And Main Contributions:**

This work proposed to generate Visual Question-Answering style pairs (Multi30k-VQA dataset) from source text via LLMs to model the probing signal in MMT explicitly. Experimental results on two representative benchmarks show the proposed MMT-VQA framework's effectiveness.

**Reasons To Accept:**

• This work released a Multi30k-VQA dataset consisting of question-answer pairs.
• This work proposed an MMT-VQA Multi-Task Learning Framework to explicitly model the probing signal in MMT to strengthen the interactions between text and visual modalities.
- Experiments and analysis prove the effectiveness of the proposed framework.

**Reasons To Reject:**

-

**Reproducibility:**

4: Could mostly reproduce the results, but there may be some variation because of sample variance or minor variations in their interpretation of the protocol or method.

**Reviewer Confidence:**

1: Not my area, or paper was hard for me to understand. My evaluation is just an educated guess.

---

> ### Author Rebuttal · Authors · 2023-08-29
>
> Thanks for your comments. If you have any other concerns toward this paper, we are glad to make a further discussion.

---

### Official Review · Reviewer_EZME · 2023-08-10

**Typos Grammar Style And Presentation Improvements:** Please proofread the manuscript.
**Soundness:** 3

**Excitement:**

3: Ambivalent: It has merits (e.g., it reports state-of-the-art results, the idea is nice), but there are key weaknesses (e.g., it describes incremental work), and it can significantly benefit from another round of revision. However, I won't object to accepting it if my co-reviewers champion it.

**Missing References:**

[c] Yi Li et al. VALHALLA: Visual Hallucination for Machine Translation. CVPR 2022

[d] Junjie Ye et al. Noise-robust Cross-modal Interactive Learning with Text2Image Mask for Multi-modal Neural Machine Translation. COLING 2022

**Paper Topic And Main Contributions:**

The main contribution of this paper is the generation of QA pairs according to the source sentences through LLM API, followed by the incorporation of probing signals into MMT, aiming to cross-modal interaction. The claim of the paper is supported by a comprehensive set of detailed experiments and analyses.

**Questions For The Authors:**

See [Reasons To Reject]. Additional suggestion:
- The colors used in Figure 5 are difficult to read.

**Reasons To Accept:**

- This paper is easy to follow and well-written.
- The introduction of QA pairs and the design of the model architecture are reasonable and intuitive.
- The analysis of the experiments conducted is thorough.

**Reasons To Reject:**

- The novelty of the paper seems limited in my opinion. As the authors claimed, the main contribution lies in the probing task proposed in [a] for training, it should be noted that the practice of incorporating additional signals for training is not new. The whole paper can be seen as its follow-up incremental work.
- The paper would benefit from including missing statistics of the constructed QA dataset.
- In the prompt design phase, there is a lack of further exploration of two examples for in-context learning, which seems to be essential for the quality of the final synthetic data.
- The paper lacks some comparisons to state-of-the-art methods, please see [Missing References].
- It is difficult to ascertain whether the improvement in performance primarily stems from better visual features or the additionally constructed QA pairs. It would be beneficial to include an ablation study to analyze the contribution of each component.
- It is recommended to provide more comparisons with close work [b] that address the same problem, to clarify the merits of the approach.
- The assessment provided in the paper is not comprehensive. Please include the widely adopted METEOR metric in the main tables as well.
- The results presented in the paper lack statistical significance. It is unclear whether they are reported for a single trial or averaged over multiple trials. Additionally, the tolerance scope is not mentioned. These details are not reflected in the paper (maybe I missed them, please clarify).
- In section 5.3, there is a lack of intuitive evidence for cross-modal interactions, as only quantitative analysis is currently provided. It is recommended to include additional qualitative evidence or visualizations to support and enhance the understanding of the cross-modal interactions.
- Contribution claims are unclear and inconsistent. The authors mention better visual features are beneficial for MMT in Line85-87, which seems to be separated from the previous introduction; in addition, the author shows better results of MAE and CLIP in Table 1, while Line85-87 mentions MAE, BLIP, this is confusing.

[a] Bei Li et al. On Vision Features in Multimodal Machine Translation. ACL 2022

[b] Baijun Ji et al. Increasing Visual Awareness in Multimodal Neural Machine Translation from an Information Theoretic Perspective. EMNLP 2022

**Reproducibility:**

4: Could mostly reproduce the results, but there may be some variation because of sample variance or minor variations in their interpretation of the protocol or method.

**Reviewer Confidence:**

4: Quite sure. I tried to check the important points carefully. It's unlikely, though conceivable, that I missed something that should affect my ratings.

---

> ### Author Rebuttal · Authors · 2023-08-29
>
> Thank you for your thoughtful comments and the time you've invested in reviewing our paper. We highly value your feedback and wish to address each of your concerns in detail.
>
> 1. **Novelty and Contributions of the Paper**: We appreciate the opportunity to clarify the unique aspects of our work. While we acknowledge that incorporating additional signals for training is not a new concept, the novelty of our work resides in the specific design and utility of probing signals during the training phase.
>
>    - **Effective Probing Signals**: Our probing signals are designed to assess the efficacy of Multimodal Translation (MMT) systems in leveraging visual patterns, drawing inspiration from previous work (Caglayan et al., 2019; Li et al., 2022). However, what sets our work apart is the application of these probing signals to actively enhance the interaction between visual and textual modalities—a line of inquiry that has not been previously explored. We believe that the incorporation of these signals into a multi-task learning framework, while straightforward, yields robust results.
>    - **Multi30k-VQA Dataset**: Another noteworthy contribution is the Multi30K-VQA dataset that we intend to release. This dataset comprises high-quality, task-specific question-answer pairs in a text-image setting, and aims to provide a fertile ground for subsequent research. We envision this dataset as a valuable resource for the community, potentially inspiring future works in this space.
>
>    We believe that these elements, taken together, distinguish our work from incremental advancements in the field and provide meaningful contributions to the research community.
>
> 2. **Missing Statistics of QA Dataset**: You're correct that comprehensive statistics of our QA dataset would strengthen the paper. We give the detailed statistic below:
>
>    As is shown in the table. We collect the Multi30k-VQA data using the strong power of LLMs. As depicted in Figure 2, our prompt design for guiding the LLM consists of three key components:
>
>    - **Task Description**: Initially, we provide an explicit task description to make the LLM aware of the intended output, focusing on generating high-quality question-answer pairs relevant to the task at hand.
>    - **Constraints and Requirements**: Following the task description, we impose specific requirements to minimize the chance of hallucination and to direct the model towards generating the intended output format, including nouns, topic numbers, colors and characters. Note that color and characters are two more representative samples for LLM to ask questions than the other two types.
>    - **Two-Shot Demonstrations**: Lastly, we supply the LLM with two demonstration examples that serve as a template for the expected format of question-answer pairs.
>
>    The resulting corpus contains 29,000 unique question-answer pairs, each generated according to our specifications. While we have designed four types of questions for this task, we intentionally provide only two simpler examples (color and characters) in the prompt. This strategic choice is made to prevent the LLM from disregarding simpler question formats and solely generating complex ones, irrespective of their suitability for a given context. Overall, the prompt here has been carefully tuned, and the output is the optimal choice as far as we tested among various prompts. The percentage of the four types are:
>
>    | Type      | Count |
>    | --------- | ----- |
>    | Noun      | 5133  |
>    | Character | 18423 |
>    | Color     | 5303  |
>    | Number    | 141   |
>
>
>
> 3. **Prompt Design for In-Context Learning**: We understand the need for deeper exploration here.  Actually, as we emphasized above, both the prompt and the demonstration have been carefully designed and varified. We found there is no significant improvement when we add more demonstrations whatever from the number or the question types. In the revised paper, we will provide further examples and discuss how these choices affect the quality of the final synthetic data.
>
> 4. **Comparison to State-of-the-Art Methods**: We appreciate this comment and will add the missing references as well as comparative evaluations with state-of-the-art methods[1] [2] to substantiate our contributions. Here, we want to do a detailed comparison between our MMT-VQA with these two related work.
>
>    - VALHALLA: This work introduced a visual hallucination framework called VALHALLA, which only requires the source sentence for reasoning and employs hallucinated visual representations for multimodal machine translation. Specifically, given a source sentence, an autoregressive hallucination transformer is used to predict a discrete visual representation from the input text, and this is then combined with the text and hallucinated representations to obtain the target translation. VALHALLA achieves significant BLEU improvements and we think our Multi30K-VQA can also bring additional benefits for theirs framework.
>    - Noise-robust Cross-modal Interactive learning: This work mainly aim to reduce the noise interference within in the image via mask strategy. The method achieves quite strong performance, but the METEOR is somehow strange compared with previous work. We also include these results in the following table.
>    - Mutual information perspective: This work mainly used mutual information to quantify the source specific-information and the target-specific information and proposed two strategies to better utilize the visual signals. The core motivation is to increase the visual awareness since MMT system concentrates more on the textual input. The main framework is orthogonal to ours and we will investigate the fusion of two methods in the revised manuscript.
>
>    [1] Yi Li et al. VALHALLA: Visual Hallucination for Machine Translation. CVPR 2022
>
>    [2] Junjie Ye et al. Noise-robust Cross-modal Interactive Learning with Text2Image Mask for Multi-modal Neural Machine Translation. COLING 2022
>
>    We will add the aforementioned comparisons in the improved version. More experiments on whether MMT-VQA can work well upon these strong MMT systems would be conducted.
>
> 5. **Ablation Study**: Your point about unclear sources of performance gains is well taken. An ablation study has been already conducted which was shown in Table 4. By comparing #10 and #11, both two systems were based on the MAE-base model. The only difference is that MMT-VQA used the probing signals during training. Thus the improvements over #10 show the effectiveness of the additional VQA pairs. Moreover, Table 7 also shows that MMT systems based on both BLIP-base and MAE-base vision features can gain benefits on the special designed test set. On the other hand, we refer you to the results in Figure 5, that when the answer or the question (of the VQA pairs generated by LLM) is wrong, our model suffers from BLEU degradation. This also indicates the training signals provided by the additional VQA pairs are indeed helpful. We think these results can address your concerns. We also make a quantitative case study as follows:
>
>    ```
>    src:  a man talks on the phone with his feet up .
>
>    ref:  ein mann telefoniert mit hochgelegten füßen .
>
>    MMT(SA):  ein mann telefoniert mit den füßen .
>
>    MMT-VQA:  ein mann telefoniert mit hochgelegten füßen .
>    ----------------------------------------------------------------------------
>    src:  a boat with red , white and blue sails docking at a pier .
>
>    ref:  ein boot mit roten , weißen und blauen segeln dockt an einem pier an .
>
>    MMT(SA):  ein boot mit rot-blauen segeln an einem pier .
>
>    MMT-VQA:  ein boot mit roten , weißen und blauen segeln an einem pier .
>    ```
>
>    In the first case, we observed that the MMT system, despite employing a selective attention mechanism and Vision Transformer (ViT) features, faltered in the translation of the word "up" in the German context. Specifically, it omitted this word entirely. In contrast, our MMT-VQA system produced the accurate German translation "hochgelegten," which effectively captures the notion of "elevated" or "up."
>
>    Another noteworthy observation is that conventional MMT systems struggle with accurate color translation when a sentence contains multiple color descriptors. This limitation was effectively mitigated in our MMT-VQA system by the incorporation of probing signals. These signals, particularly those covering color question-answer pairs, significantly enhanced the system's ability to resolve such translation ambiguities.
>
>    Our analysis suggests that the failure of the MMT system with selective attention in translating seemingly simple sentences stems primarily from its inability to disambiguate certain words effectively.
>
> 6. **Comparison with Close Work**: We agree that a comparison with work [b] is crucial. Similar question with the Weakness 4.
>
> 7. **Comprehensive Assessment and METEOR Metric**: We understand that incorporating the METEOR metric can provide a more comprehensive assessment. We will include this in the main tables.
>
>    ## Multi30k En-De:
>
>    | #    | System              | Feature   | Test2016 BLEU | Test2016 METEOR | Test2017 BLEU | Test2017 METEOR | MSCOCO BLEU | MSCOCO METEOR | Test2018 BLEU | Test2018 METEOR |
>    | ---- | ------------------- | --------- | ------------- | --------------- | ------------- | --------------- | ----------- | ------------- | ------------- | --------------- |
>    | 1    | Transformer Tiny    | -         | 41.02         | 68.22           | 33.36         | 62.05           | 29.88       | 56.64         | 30.52         | 57.37           |
>    | 2    | Doubly-ATT          | ResNet    | 41.45         | 68.04           | 33.95         | 61.83           | 29.63       | 59.21         | -             | -               |
>    | 3    | Imagination         | ResNet    | 41.31         | 68.06           | 32.89         | 61.29           | 29.90       | 56.57         | -             | -               |
>    | 4    | UVR-NMT             | ResNet    | 40.79         | -               | 32.16         | -               | 29.02       | -             | -             | -               |
>    | 5    | Gated Fusion        | ResNet    | 41.96         | 67.84           | 33.59         | 61.94           | 29.04       | 56.15         | -             | -               |
>    | 6    | Selective Attention | ViT-base  | 41.93         | 68.55           | 33.60         | 61.42           | 31.14       | 56.77         | -             | -               |
>    | 7    | IKD-MMT             | ResNet    | 41.28         | 58.93           | 33.83         | 53.21           | 30.17       | 48.93         | -             | -               |
>    | 8    | PLUVR               | ResNet    | 40.30         | -               | 33.45         | -               | 30.28       | -             | -             | -               |
>    | 9    | IVA                 | ResNet    | 41.77         | 68.60           | 34.58         | 62.40           | 30.61       | 56.70         | -             | -               |
>    | 10   | VALHALLA            | VQGAN VAE | 42.60         | 69.30           | 35.10         | 62.80           | 30.70       | 57.60         | -             | -               |
>    | 11   | Noise-robust        | Resnet    | 42.56         | 59.98           | 35.09         | 54.51           | 31.09       | 50.46         | -             | -               |
>    | 12   | Multimodal-mixup    | Resnet    | 41.77         | 58.93           | 33.07         | 51.85           | 29.90       | 49.09         | -             | -               |
>    | 13   | Selective Attention | ViT-base  | 40.68         | 67.80           | 33.35         | 61.00           | 28.61       | 55.30         | 30.41         | 57.29           |
>    | 14   | Selective Attention | MAE-base  | 41.65         | 68.43           | 34.27         | 62.08           | 29.75       | 56.68         | 31.09         | 58.17           |
>    | 15   | MMT-VQA             | MAE-base  | 42.55         | 69.00           | 34.58         | 61.99           | 29.94       | 57.23         | 31.74         | 58.49           |
>
>    ## Multi30k En-Fr:
>
>    | #    | System              | Feature   | Test2016 BLEU | Test2016 METEOR | Test2017 BLEU | Test2017 METEOR | MSCOCO BLEU | MSCOCO METEOR | Test2018 BLEU | Test2018 METEOR |
>    | ---- | ------------------- | --------- | ------------- | --------------- | ------------- | --------------- | ----------- | ------------- | ------------- | --------------- |
>    | 1    | Transformer Tiny    | -         | 61.80         | 81.02           | 53.46         | 75.62           | 44.52       | 69.43         | 38.05         | 64.01           |
>    | 2    | Doubly-ATT          | ResNet    | 61.99         | 81.12           | 53.72         | 75.71           | 45.16       | 70.25         | -             | -               |
>    | 3    | Imagination         | ResNet    | 61.90         | 81.20           | 54.07         | 76.03           | 44.81       | 70.35         | -             | -               |
>    | 4    | UVR-NMT             | ResNet    | 61.00         | -               | 53.20         | -               | 43.71       | -             | -             | -               |
>    | 5    | Gated Fusion        | ResNet    | 61.69         | 80.97           | 54.85         | 76.34           | 44.86       | 70.51         | -             | -               |
>    | 6    | Selective Attention | ViT-base  | 62.48         | 81.71           | 54.44         | 76.46           | 44.72       | 71.20         | -             | -               |
>    | 7    | IKD-MMT             | ResNet    | 62.53         | 77.20           | 54.84         | 71.87           | -           | -             | -             | -               |
>    | 8    | PLUVR               | ResNet    | 61.31         | -               | 53.15         | -               | 43.65       | -             | -             | -               |
>    | 9    | IVA                 | ResNet    | -             | -               | -             | -               | -           | -             | -             | -               |
>    | 10   | VALHALLA            | VQGAN VAE | 63.10         | 81.80           | 56.00         | 77.10           | 46.40       | 71.30         | -             | -               |
>    | 11   | Noise-robust        | Resnet    | 63.24         | 77.54           | 55.48         | 72.62           | 46.34       | 67.40         | -             | -               |
>    | 12   | Multimodal-mixup    | Resnet    | 62.23         | 76.85           | 55.18         | 73.37           | 44.42       | 66.41         | -             | -               |
>    | 13   | Selective Attention | ViT-base  | 61.65         | 80.96           | 54.13         | 76.06           | 44.40       | 70.48         | 38.10         | 64.17           |
>    | 14   | Selective Attention | MAE-base  | 61.87         | 80.76           | 54.07         | 76.10           | 44.65       | 70.43         | 38.18         | 64.55           |
>    | 15   | MMT-VQA             | MAE-base  | 62.24         | 81.77           | 54.89         | 76.53           | 45.75       | 71.21         | 38.72         | 64.87           |
>
> 8. **Statistical Significance**: The results were averaged over three trials and include tolerance scopes to ensure the statistical robustness of our results. We both run the models in seed 1, 42 and 2023, respectively. Thanks for pointing out this important issue. We will carefully revise the experimental setup to avoid the unnecessary misunderstandings!
>
> 9. **Intuitive Evidence for Cross-Modal Interactions**: We think this is a similar concern with 5. The provided two cases may address your concern well. Moreover, we do other analyses from three aspects:
>
>    - **From the Perspective of Part-of-Speech (POS):** We analyzed translation quality using POS tagging. For each POS type in the hypothesis, we counted its occurrence in sentences and then segmented the test set accordingly. We compared BLEU scores for the MMT-VQA and SA methods across these subsets. To isolate the effect of specific POS types, we calculated the score differences and adjusted for word frequency. Our results highlight that adjectives (JJ) and plural nouns (NNS) most significantly improve translation quality when using the VQA method.
>
>    - **From the Perspective of Long and Difficult Words:** We found that 3% of vocabulary consists of long and difficult words. After calculating that the 97th percentile of all word lengths is 9, we divided the test set based on whether sentences contained long and difficult words. We found that the MMT-VQA method improved the BLEU score by 0.98 points on the test subset with long and difficult words compared to the selective attention baseline method (Li et al., 2022 ). The improvement on the subset without long and difficult words was 0.67 points.
>
>    - **From the Perspective of Long and Difficult Sentences:** We then completed the identification of long and difficult sentences. We first determined the number of clusters for sentence length categorization to be 2 based on the elbow method, dividing sentence lengths into two categories: long sentences and short sentences. We found that during the translation of long sentences, the BLEU score difference between the VQA and SA methods was 1.08, and for short sentences, it was 0.63—nearly a 70% improvement.
>
> 10. **Clarity of Contribution Claims**: We apologize for any confusion generated by the inconsistencies in Lines 85-87. Our contributions are primarily two-fold:
>
>     - **Advanced Visual Features**: Our work delves into the effectiveness of advanced visual features, particularly in the context of MMT systems built on Vision Transformers (ViT). In doing so, we expand upon the research initiated by Li et al., 2022. Although one might intuitively expect that CLIP and BLIP would outperform MAE and ViT after optimization on extensive text-image paired data, our findings indicate otherwise. This observation serves as an invaluable guide for future research in the field, including the study of multimodal Language Models (LLMs).
>     - **Probing Signals**: While probing tasks and other visual-grounded test sets have garnered significant attention, the unanswered question remains: can these evaluative measures be used to actively strengthen MMT systems? In our work, we focus on leveraging these probing signals to enhance the interaction between visual and textual modalities during training. Although we concede that multi-task learning is a well-established method, it serves as an effective conduit to achieve our research goals.
>
> We hope the aforementioned responses can well addressed your concerns, and we really appreciate for your valuable feedbacks for further improving the quality of our paper. Wish you can have another turn to make a new evaluation towards our paper.

---

### Official Review · Reviewer_ntqX · 2023-08-10

**Soundness:** 2

**Excitement:**

2: Mediocre: This paper makes marginal contributions (vs non-contemporaneous work), so I would rather not see it in the conference.

**Missing References:**

[1] Yi Li, Rameswar Panda, Yoon Kim, Chun-Fu Richard Chen, Rogerio S Feris, David Cox, and Nuno Vasconcelos. 2022. VALHALLA: Visual Hallucination for Machine Translation. In Proceedings of the IEEE/CVF Conference on Computer Vision and Pattern Recognition. 5216–5226.

[2] Junjie Ye, Junjun Guo, Yan Xiang, Kaiwen Tan, and Zhengtao Yu. 2022. Noiserobust Cross-modal Interactive Learning with Text2Image Mask for Multi-modal Neural Machine Translation. In Proceedings of the 29th International Conference on Computational Linguistics. 5098–5108.

[3] Junjie Ye and Junjun Guo. 2022. Dual-level interactive multimodal-mixup encoder for multi-modal neural machine translation. Applied Intelligence 52, 12 (2022), 14194–14203.

**Paper Topic And Main Contributions:**

The paper release a Multi30k-VQA dataset via using LLMs API and propose a MMT-VQA Multi-Task Learning Framework to explicitly model the probing signal in MMT.

**Questions For The Authors:**

Could you give a case study to show the superiority of your method?

**Reasons To Accept:**

1. They introduce LLM to MMT, which is interesting.
2. Experimental results on Multi30K En-De and En-Fr demonstrate the effectiveness both in terms of BLEU and specific test sets.
3. The figures are clear and beneficial for readers to understand the paper.

**Reasons To Reject:**

1. The paper don't compare their models with more recent SOTAs [1-3], so it can not get higher soundness.
2. You should provide the results on more datasets, such as Test2018.
3. You should provide the METEOR results, which is also reported in recent works [1-5].
4. The Figure 5 is not clear, you should give more explanation about it.

[1] Yi Li, Rameswar Panda, Yoon Kim, Chun-Fu Richard Chen, Rogerio S Feris, David Cox, and Nuno Vasconcelos. 2022. VALHALLA: Visual Hallucination for Machine Translation. In Proceedings of the IEEE/CVF Conference on Computer Vision and Pattern Recognition. 5216–5226.

[2] Junjie Ye, Junjun Guo, Yan Xiang, Kaiwen Tan, and Zhengtao Yu. 2022. Noiserobust Cross-modal Interactive Learning with Text2Image Mask for Multi-modal Neural Machine Translation. In Proceedings of the 29th International Conference on Computational Linguistics. 5098–5108.

[3] Junjie Ye and Junjun Guo. 2022. Dual-level interactive multimodal-mixup encoder for multi-modal neural machine translation. Applied Intelligence 52, 12 (2022), 14194–14203.

[4]  Good for Misconceived Reasons: An Empirical Revisiting on the Need for Visual Context in Multimodal Machine Translation. In Proceedings of the 59th Annual Meeting of the Association for Computational Linguistics and the 11th International Joint Conference on Natural Language Processing (Volume 1: Long Papers). 6153–6166.

[5] Li B, Lv C, Zhou Z, et al. On Vision Features in Multimodal Machine Translation[C]//Proceedings of the 60th Annual Meeting of the Association for Computational Linguistics (Volume 1: Long Papers). 2022: 6327-6337.

**Reproducibility:**

2: Would be hard pressed to reproduce the results. The contribution depends on data that are simply not available outside the author's institution or consortium; not enough details are provided.

**Reviewer Confidence:**

4: Quite sure. I tried to check the important points carefully. It's unlikely, though conceivable, that I missed something that should affect my ratings.

---

> ### Author Rebuttal · Authors · 2023-08-29
>
> Thanks for your comments on our paper. We think all the issues would be well addressed in our improved version.
>
> Here are the detailed response for your main concerns:
>
> - We will add the comparison with the pointed missing reference. Thanks for pointing out this issue. And we have already included these results in the following table.
>
> - Since most of previous work reported the results on test2016, test2017 and MSCOCO. Towards a fair comparison with theirs,  we only report results on these three testsets. For a more convincing result, we further conducted experiments on Test2018.
>
>   | System              | Feature | Test2018 (BLEU) | Test2018 (METEOR) |
>   | ------------------- | ------- | --------------- | ----------------- |
>   | Transformer-tiny    | -       | 30.52           | 57.37             |
>   | Selective Attention | ViT     | 30.41           | 57.29             |
>   | Selective Attention | MAE     | 31.09           | 58.17             |
>   | MMT-VQA             | MAE     | 31.74           | 58.49             |
>
>   As is shown in the table, our MMT-VQA can achieve strong performance on test2018 in terms of both BLEU and METEOR
>
> - Thanks for pointing out the missing meteor scores and we show them as below:
>
>   | #    | System              | Feature   | Test2016 BLEU | Test2016 METEOR | Test2017 BLEU | Test2017 METEOR | MSCOCO BLEU | MSCOCO METEOR | Test2018 BLEU | Test2018 METEOR |
>   | ---- | ------------------- | --------- | ------------- | --------------- | ------------- | --------------- | ----------- | ------------- | ------------- | --------------- |
>   | 1    | Transformer Tiny    | -         | 41.02         | 68.22           | 33.36         | 62.05           | 29.88       | 56.64         | 30.52         | 57.37           |
>   | 2    | Doubly-ATT          | ResNet    | 41.45         | 68.04           | 33.95         | 61.83           | 29.63       | 59.21         | -             | -               |
>   | 3    | Imagination         | ResNet    | 41.31         | 68.06           | 32.89         | 61.29           | 29.90       | 56.57         | -             | -               |
>   | 4    | UVR-NMT             | ResNet    | 40.79         | -               | 32.16         | -               | 29.02       | -             | -             | -               |
>   | 5    | Gated Fusion        | ResNet    | 41.96         | 67.84           | 33.59         | 61.94           | 29.04       | 56.15         | -             | -               |
>   | 6    | Selective Attention | ViT-base  | 41.93         | 68.55           | 33.60         | 61.42           | 31.14       | 56.77         | -             | -               |
>   | 7    | IKD-MMT             | ResNet    | 41.28         | 58.93           | 33.83         | 53.21           | 30.17       | 48.93         | -             | -               |
>   | 8    | PLUVR               | ResNet    | 40.30         | -               | 33.45         | -               | 30.28       | -             | -             | -               |
>   | 9    | IVA                 | ResNet    | 41.77         | 68.60           | 34.58         | 62.40           | 30.61       | 56.70         | -             | -               |
>   | 10   | VALHALLA            | VQGAN VAE | 42.60         | 69.30           | 35.10         | 62.80           | 30.70       | 57.60         | -             | -               |
>   | 11   | Noise-robust        | Resnet    | 42.56         | 59.98           | 35.09         | 54.51           | 31.09       | 50.46         | -             | -               |
>   | 12   | Multimodal-mixup    | Resnet    | 41.77         | 58.93           | 33.07         | 51.85           | 29.90       | 49.09         | -             | -               |
>   | 13   | Selective Attention | ViT-base  | 40.68         | 67.80           | 33.35         | 61.00           | 28.61       | 55.30         | 30.41         | 57.29           |
>   | 14   | Selective Attention | MAE-base  | 41.65         | 68.43           | 34.27         | 62.08           | 29.75       | 56.68         | 31.09         | 58.17           |
>   | 15   | MMT-VQA             | MAE-base  | 42.55         | 69.00           | 34.58         | 61.99           | 29.94       | 57.23         | 31.74         | 58.49           |
>
> - Our experiments reveal that the model is notably sensitive to the quality of question-answer pairs within the Multi30k-VQA dataset. Specifically, when a correct question is erroneously paired with an incorrect answer, the model exhibits a marked decline in performance after being trained on such a flawed dataset. This observation necessitates a cautious approach in creating high-quality datasets for training. We will restructure this section to provide a more comprehensive understanding of this critical dependency, thereby aiding in the interpretation and generalization of our results.

---

### Meta-Review · Area_Chair_Yjp6 · 2023-09-11

**Recommendation:** 3

**Metareview:**

The paper works on the MMT problems. It focuses on enhancing the multimodal interactions to achieve a better results. The proposed method is to connect MMT and VQA. Overall, the paper achieves both technical novelty (might be limited due to the reviewer's feedback) and empirical improvements.

A concern from the reviewing process is that the paper does not include the results from some 2022 papers. It's hard to fully stand with author's point that these results are just "missing references", since some of the paper achieves higher results. Higher previous results does not directly change the paper's evaluation, but it might change the claims made with this paper.

It would be great if the author can have an update over the paper claims as well (e.g., the claim of "state-of-the-art" in abstract; I want to mention once more that not claiming SotA would not hurt the paper performance, but a claim with uncertainty could). Thus it can resolve confusions for potential readers.

---

### Decision · Program_Chairs · 2023-10-07

**Decision:**

Accept-Findings

**Comment:**

The paper works on the MMT problems. It focuses on enhancing the multimodal interactions to achieve a better results. The proposed method is to connect MMT and VQA. Overall, the paper achieves both technical novelty (might be limited due to the reviewer's feedback) and empirical improvements.

A concern from the reviewing process is that the paper does not include the results from some 2022 papers. It's hard to fully stand with author's point that these results are just "missing references", since some of the paper achieves higher results. Higher previous results does not directly change the paper's evaluation, but it might change the claims made with this paper.

It would be great if the author can have an update over the paper claims as well (e.g., the claim of "state-of-the-art" in abstract; I want to mention once more that not claiming SotA would not hurt the paper performance, but a claim with uncertainty could). Thus it can resolve confusions for potential readers.